# Robot Parkour Learning

Ziwen Zhuang[*13] Zipeng Fu[*2] Jianren Wang[4] Christopher Atkeson[4] Sören Schwertfeger[3]
Chelsea Finn[2] Hang Zhao[15]

[1]Shanghai Qi Zhi, [2]Stanford, [3]ShanghaiTech, [4]CMU, [5]Tsinghua, [*]project co-leads
project website:  https://robot-parkour.github.io

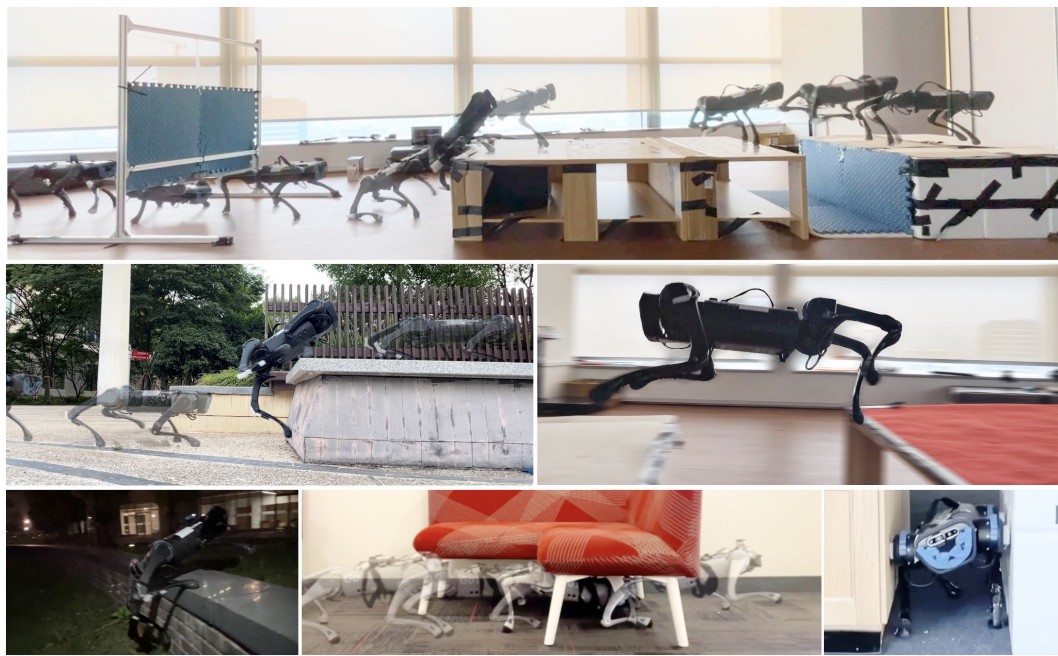

Figure 1: We present a framework for learning parkour skills on low-cost robots. Our end-to-end vision-based parkour learning system enable the robot to climb high obstacles, leap over large gaps, crawl beneath low barriers, squeeze through thin slits and run. Videos are on the project website.

**Abstract:** Parkour is a grand challenge for legged locomotion that requires robots to overcome various obstacles rapidly in complex environments. Existing methods can generate either diverse but blind locomotion skills or vision-based but specialized skills by using reference animal data or complex rewards. However, *autonomous* parkour requires robots to learn generalizable skills that are both vision-based and diverse to perceive and react to various scenarios. In this work, we propose a system for learning a single end-to-end vision-based parkour policy of diverse parkour skills using a simple reward without any reference motion data. We develop a reinforcement learning method inspired by direct collocation to generate parkour skills, including climbing over high obstacles, leaping over large gaps, crawling beneath low barriers, squeezing through thin slits, and running. We distill these skills into a single vision-based parkour policy and transfer it to a quadrupedal robot using its egocentric depth camera. We demonstrate that our system can empower two different low-cost robots to autonomously select and execute appropriate parkour skills to traverse challenging real-world environments.

**Keywords:** Agile Locomotion, Visuomotor Control, Sim-to-Real Transfer

## 1  Introduction

Humans and animals possess amazing athletic intelligence.  Parkour is an examplar of athletic intelligence of many biological beings capable of moving swiftly and overcoming various obstacles in complex environments by running, climbing, and jumping [1]. Such agile and dynamic movements

7th Conference on Robot Learning (CoRL 2023), Atlanta, USA.

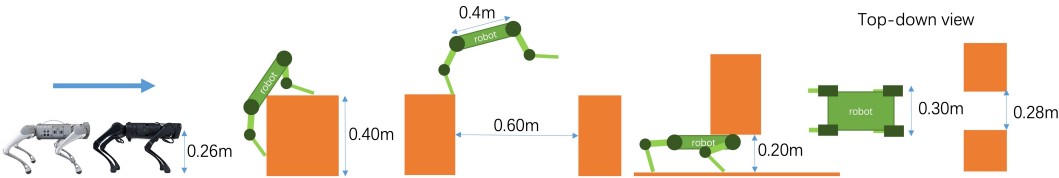

Figure 2: We illustrate the challenging obstacles that our system can solve, including climbing high obstacles of 0.40m (1.53x robot height), leap over large gaps of 0.60m (1.5x robot length), crawling beneath low barriers of 0.2m (0.76x robot height), squeezing through thin slits of 0.28m by tilting (less than the robot width).

require real-time visual perception and memorization of surrounding environments [2, 3], tight coupling of perception and action [4, 5], and powerful limbs to negotiate barriers [6]. One of the grand challenges of robot locomotion is building autonomous parkour systems.

Boston Dynamics Atlas robots [7] have demonstrated stunning parkour skills. However, the massive engineering efforts needed for modeling the robot and its surrounding environments for predictive control and the high hardware cost prevent people from reproducing parkour behaviors given a reasonable budget. Recently, learning-based methods have shown robust performance on walking [8, 9, 10, 11, 12, 13, 14, 15, 16, 17, 18, 19, 20, 12, 21, 22, 23, 24, 25, 26, 27], climbing stairs [20, 28, 29, 30, 31, 32, 33], mimicking animals [34, 35, 36, 37, 38, 39] and legged mobile manipulation [40, 41, 42] by learning a policy in simulation and transferring it to the real world while avoiding much costly engineering and design needed for robot-specific modeling. Can we leverage learning-based methods for robot parkour but only using low-cost hardware?

There are several challenges for robot parkour learning. First, learning diverse parkour skills (e.g. running, climbing, leaping, crawling, squeezing through, and etc) is challenging. Existing reinforcement learning works craft complex reward functions of many terms to elicit desirable behaviors of legged robots. Often each behavior requires manual tuning of the reward terms and hyper-parameters; thus these works are not scalable enough for principled generation of a wide range of agile parkour skills. In contrast, learning by directly imitating animals' motion capture data can circumvent tedious reward design and tuning [34, 43], but the lack of egocentric vision data and diverse animal MoCap skills prevents the robots from learning diverse agile skills and autonomously selecting skills by perceiving environment conditions. Second, obstacles can be challenging for low-cost robots of small sizes, as illustrated in Figure 2. Third, beyond the challenge of learning diverse skills, visual perception is dynamical and laggy during high-speed locomotion. For example, when a robot moves at 1m/s, a short 0.2 second of signal communication delay will cause a perception discrepancy of 0.2m (7.9 inches). Existing learning-based methods have not demonstrated effective high-speed agile locomotion. Lastly, parkour drives the electric motors to their maximum capacity, so proactive measures to mitigate potential damage to the motors must be included in the system.

This paper introduces a robot parkour learning system for low-cost quadrupedal robots that can perform various parkour skills, such as climbing over high obstacles, leaping over large gaps, crawling beneath low barriers, squeezing through thin slits, and running. Our reinforcement learning method is inspired by direct collocation and consists of two simulated training stages: RL pre-training with soft dynamics constraints and RL fine-tuning with hard dynamics constraints. In the RL pre-training stage, we allow robots to penetrate obstacles using an automatic curriculum that enforces soft dynamics constraints. This encourages robots to gradually learn to overcome these obstacles while minimizing penetrations. In the RL fine-tuning stage, we enforce all dynamics constraints and fine-tune the behaviors learned in the pre-training stage with realistic dynamics. In both stages, we only use a simple reward function that motivates robots to move forward while conserving mechanical energy. After each individual parkour skill is learned, we use DAgger [44, 45] to distill them into a single vision-based parkour policy that can be deployed to a legged robot using only onboard perception and computation power.

The main contributions of this paper include:

- **an open-source system for robot parkour learning**, offering a platform for researchers to train and deploy policies for agile locomotion;
- **a two-stage RL method** for overcoming difficult exploration problems, involving a pre-training stage with soft dynamics constraints and a fine-tuning stage with hard dynamics constraints;

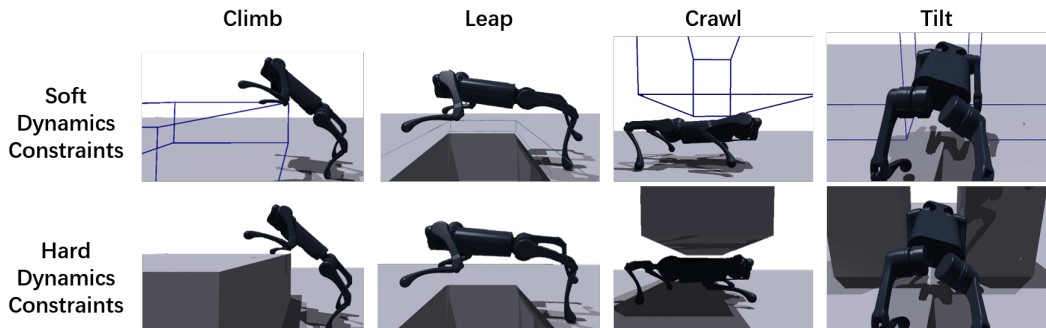

Figure 3: Soft dynamics constraints and hard dynamics constraints for each skill. Given soft dynamics constraints, the obstacles are penetrable.

- **extensive experiments in simulation and the real world** showing that our parkour policy enables low-cost quadrupedal robots to autonomously select and execute appropriate parkour skills to traverse challenging environments in the open world using only onboard computation, onboard visual sensing and onboard power, including climbing high obstacles of 0.40m (1.53x robot height), leap over large gaps of 0.60m (1.5x robot length), crawling beneath low barriers of 0.2m (0.76x robot height), squeezing through thin slits of 0.28m by tilting (less than the robot width), and running;
- **generalization to different robots**, where we demonstrate that our system with the same training pipeline can power two different robots, A1 and Go1.

## 2 Related Work

**Agile Locomotion.** Model-based control has achieved much success in agile locomotion, from MIT Cheetah robots and A1 robots jumping over or onto obstacles of various heights [46, 47, 48], ETH StarlETH robots jumping vertically [49], CMU Unified Snake robots climbing trees [50], X-RHex robots self-righting using tails [51], ANYmal ALMA robots opening doors [52], ATRIAS robots walking over stepping stones [53, 54], Marc Raibert's One-Legged Hopping Machine [55], and Boston Dynamics Atlas' parkour skills [7]. Recently, learning-based methods have also demonstrated various agile locomotion capabilities, including high-speed running [56, 16, 57, 35], resetting to the standing pose from random states [11, 38, 15, 58], jumping [59, 60, 61], climbing stairs [20, 10, 28, 29, 30, 32, 33], climbing over obstacles [62], walking on stepping stones [29], back-flipping [63], quadrupedal standing up on rear legs [43], opening doors [40, 64, 65, 66], moving with damaged parts [67], catching flying objects [68], balancing using a tail [69], playing football/soccer [70, 71, 72, 73], weaving through poles [74] and climbing ramps [74]. Most of these skills are blind or rely on state estimation, and specialized methods are designed for these individual skills. In contrast, we build a system for learning a single end-to-end vision-based parkour policy for various parkour skills.

**Vision-Based Locomotion.** Classical modular methods rely on decoupled visual perception and control pipelines, where the elevation maps [75, 76, 77, 78, 79, 80, 81], traversability maps [82, 83, 84, 85], or state estimators [86, 87, 88, 89, 90, 91, 92, 93, 94, 95] are constructed as intermediate representations for downstream foothold planning, path planning and control [96, 97, 98, 99, 100, 101, 102, 103, 104, 105, 106, 107, 108]. Recently, end-to-end learning-based methods have also incorporated visual information into locomotion, where visual perception is performed using depth sensing [29, 61, 31], elevation maps [28, 109, 110, 111, 112], lidar scans [113], RGB images [32], event cameras [68] or learned neural spaces [30, 33], but none have demonstrated effective high-speed agile locomotion.

## 3 Robot Parkour Learning Systems

Our goal is to build an end-to-end parkour system that directly uses raw onboard depth sensing and proprioception to control every joint of a low-cost robot to perform various agile parkour skills, such as climbing over high obstacles, leaping over large gaps, crawling beneath low barriers, squeezing through thin slits, and running. Unlike prior work where different methods and training schemes are used for different locomotion skills, we aim to generate these five parkour skills automatically and systemically. To achieve this, we develop a two-stage reinforcement learning method that is

inspired by direct collocation to learn these parkour skills under the same framework. In the RL pre-training stage, we allow robots to penetrate obstacles using an automatic curriculum that enforces soft dynamics constraints. We encourage robots to gradually learn to overcome these obstacles while minimizing penetrations and mechanical energy. In the RL fine-tuning stage, we fine-tune the pre-trained behaviors with realistic dynamics. In both stages, we only use a simple reward function that motivates robots to move forward while conserving mechanical energy. After each individual parkour skill is learned, we use DAgger [44, 45] to distill them into a single vision-based parkour policy that can be deployed. For robust sim-to-real deployment on a low-cost robot, we employ several pre-processing techniques for the depth images, calibrate onboard visual delays, and enforce proactive measures for motor safety.

## 3.1 Parkour Skills Learning via Two-Stage RL

Since depth images are costly to render, and directly training RL on visual data is not always stable, we use privileged visual information about the environments to help RL to generate specialized parkour skills in simulation. The privileged visual information includes the distance from the robot's current position to the obstacle in front of the robot, the height of the obstacle, the width of the obstacle, and a 4-dimensional one-hot category representing the four types of obstacles. We formulate each specialized skill policy as a gated recurrent neural network (GRU [114]). The inputs to a policy other than the recurrent latent state are proprioception

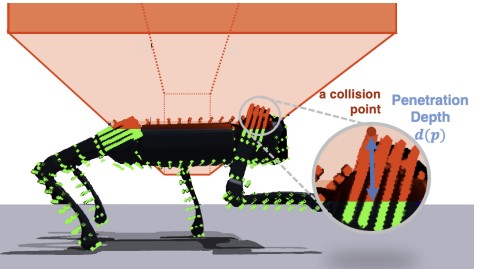

Figure 4: We show collisions points on the robot. Collision points that penetrate obstacles are in red.

$s_t^{\text{proprio}} \in \mathbb{R}^{29}$ (row, pitch, base angular velocities, positions and velocities of joints), last action $a_{t-1} \in \mathbb{R}^{12}$, the privileged visual information $e_t^{\text{vis}}$, and the privileged physics information $e_t^{\text{phy}}$. We use a similar approach to prior work [8, 10] to sample physics properties like terrain friction, center of mass of the robot base, motor strength and etc to enable domain adaptation from simulation to the real world. The policy outputs the target joint positions $a_t \in \mathbb{R}^{12}$.

We train all the specialized skill policies $\pi_{\text{climb}}, \pi_{\text{leap}}, \pi_{\text{crawl}}, \pi_{\text{tilt}}, \pi_{\text{run}}$ separately on corresponding terrains shown in Figure 3 using the same reward structure. We use the formulation of minimizing mechanical energy in [35] to derive a general skill reward $r_{\text{skill}}$ suitable for generating all skills with natural motions, which only consists of three parts, a forward reward $r_{\text{forward}}$, an energy reward $r_{\text{energy}}$ and an alive bonus $r_{\text{alive}}$:

$$r_{\text{skill}} = r_{\text{forward}} + r_{\text{energy}} + r_{\text{alive}},$$

where
$$r_{\text{forward}} = -\alpha_1 * |v_x - v_x^{\text{target}}| - \alpha_2 * |v_y|^2 + \alpha_3 * e^{-|\omega_{\text{yaw}}|},$$
$$r_{\text{energy}} = -\alpha_4 * \sum_{j \in \text{joints}} |\tau_j \dot{q}_j|^2, \quad r_{\text{alive}} = 2.$$

Measured at every time step, $v_x$ is the forward base linear velocity, $v_x^{\text{target}}$ is the target speed, $v_y$ is the lateral base linear velocity, $\omega_{\text{yaw}}$ is the base angular yaw velocity, $\tau_j$ is the torque at joint $j$, $\omega_{\text{yaw}}$ is the joint velocity at at joint $j$, and $\alpha$ are hyperparameters. We set the target speed for all skills to around 1 m/s. We use the second power of motor power at each joint to reduce both the average and the variance of motor power across all joints. See the supplementary for all hyperparameters.

**RL Pre-training with Soft Dynamics Constraints.** As illustrated in Figure 2, the difficult learning environments for parkour skills prevent generic RL algorithms from effectively finding policies that can overcome these challenging obstacles. Inspired by direct collocation with soft constraints, we propose to use soft dynamics constraints to solve these difficult exploration problems. Shown in Figure 3, we set the obstacles to be penetrable so the robot can violate the physical dynamics in the simulation by directly go through the obstacles without get stuck near the obstacles as a result of local minima of RL training with the realistic dynamics, i.e. hard dynamics

| Skill | Obstacle Properties | Training Ranges ($[l_{\text{easy}}, l_{\text{hard}}]$) | Test Ranges ($[l_{\text{easy}}, l_{\text{hard}}]$) |
|---|---|---|---|
| Climb | obstacle height | [0.2, 0.45] | [0.25, 0.5] |
| Leap | gap length | [0.2, 0.8] | [0.3, 0.9] |
| Crawl | clearance | [0.32, 0.22] | [0.3, 0.2] |
| Tilt | path width | [0.32, 0.28] | [0.3, 0.26] |

Table 1: Ranges for obstacle properties for each skill during training, measured in meters.

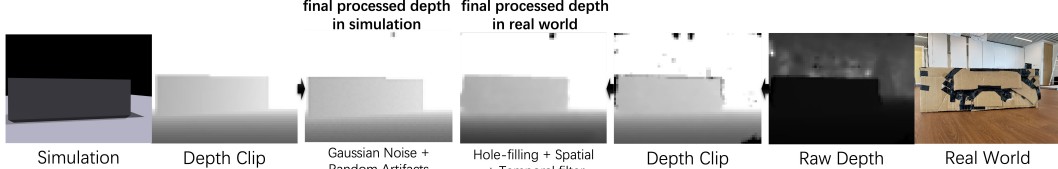

Figure 5: We bridge the visual gap between simulation and real world by applying pre-processing techniques. We use depth clipping, Gaussian noise and random artifacts in simulation, and depth clipping and hole-filling, spatial and temporal filters in the real world.

constraints. Similar to the Lagrangian formulation of direct collocation [115], we develop a penetration reward $r_{\text{penetrate}}$ to gradually enforce the dynamics constraints and an automatic curriculum that adaptively adjusts the difficulty of obstacles. This idea has also been explored in robot manipulation [116, 117]. Shown in Figure 4, to measure the degree of dynamics constraints' violation, we sample collision points within the collision bodies of the robot in order to measure the volume and the depth of penetration. Since the hips and shoulders of the robot contain all the motors, we sample more collision points around these volumes to enforce stronger dynamics constraints, encouraging fewer collisions of these vulnerable body parts in the real world. Denote a collision point on the collision bodies as $p$, an indicator function of whether $p$ violates the soft dynamics constraints as $\mathbb{1}[p]$, and the distance of $p$ to the penetrated obstacle surface as $d(p)$. The volume of penetration can be approximated by the sum of $\mathbb{1}[p]$ over all the collision points, and the average depth of penetration can be approximated by the sum of $d(p)$. In Figure 4, the collisions points violating the soft dynamics constraints ($\mathbb{1}[p] = 1$) are in red, and those with $\mathbb{1}[p] = 0$ are in green. Concretely, the penetration reward is

$$r_{\text{penetrate}} = -\sum_p \left( \alpha_5 * \mathbb{1}[p] + \alpha_6 * d(p) \right) * v_x,$$

where $\alpha_5$ and $\alpha_6$ are two fixed constants. We multiply both the penetration volume and the penetration depth with the forward base velocity $v_x$ to prevent the robot from exploiting the penetration reward by sprinting through the obstacles to avoid high cumulative penalties over time. In addition, we implement an automatic curriculum that adaptively adjusts the difficulty of the obstacles after a reset based on the performance of individual robots simulated in parallel in simulation. We first calculate the performance of a robot based on its penetration reward averaged over the previous episode before the reset. If the penetration reward is over a threshold, we increase the difficulty score $s$ of obstacles that the robot will face by one unit (0.05); if lower, then we decrease it by one unit. Every robot starts with a difficulty score 0 and the maximum difficulty score is 1. We set the obstacle property for the robot based on its difficulty score by $(1 - s) * l_{\text{easy}} + s * l_{\text{hard}}$, where $l_{\text{easy}}$ and $l_{\text{hard}}$ are the two limits of the ranges of obstacle properties corresponding to different parkour skills (shown in Table 1). We pre-train the specialized parkour skills with soft dynamics constraints using PPO [118] with the sum of the general skill reward and the penetration reward $r_{\text{skill}} + r_{\text{penetrate}}$.

**RL Fine-tuning with Hard Dynamics Constraints.** After the pre-training stage of RL is near convergence, we fine-tune every specialized parkour skill policy on the realistic hard dynamics constraints (shown in Figure 3); hence, no penetrations between the robots and obstacles are possible at the second stage of RL. We use PPO to fine-tune the specialized skills using only the general skill reward $r_{\text{skill}}$. We randomly sample obstacle properties from the ranges listed in Table 1 during fine-tuning. Since the running skill is trained on terrains without obstacles, we directly train the running skill with hard dynamics constraints and skip the RL pre-training stage with soft dynamics constraints.

### 3.2 Learning a Single Parkour Policy by Distillation

The learned specialized parkour skills are five policies that use both the privileged visual information $e_t^{\text{vis}}$, and the privileged physics information $e_t^{\text{phy}}$. However, the ground-truth privilege information is only available in the simulation but not in the real world. Furthermore, each specialized policy can only execute one skill and cannot autonomously execute and switch between different parkour skills based on visual perception of the environments. We propose to use DAgger [44, 45] to distill a single vision-based parkour policy $\pi_{\text{parkour}}$ using only onboard sensing from the five specialized skill policies $\pi_{\text{climb}}, \pi_{\text{leap}}, \pi_{\text{crawl}}, \pi_{\text{tilt}}, \pi_{\text{run}}$. We randomly sample obstacles types and properties from Table 1 to form a simulation terrain consisting of 40 tracks and 20 obstacles on each track. Since we

| | Success Rate (%) ↑ | | | | | Average Distance (m) ↑ | | | | |
|---|---|---|---|---|---|---|---|---|---|---|
| | Climb | Leap | Crawl | Tilt | Run | Climb | Leap | Crawl | Tilt | Run |
| Blind | 0 | 0 | 13 | 0 | 100 | 1.53 | 1.86 | 2.01 | 1.62 | 3.6 |
| MLP | 0 | 1 | 63 | 43 | 100 | 1.59 | 1.74 | 3.27 | 2.31 | 3.6 |
| No Distill | 0 | 0 | 73 | 0 | 100 | 1.57 | 1.75 | 2.76 | 1.86 | 3.6 |
| RMA [8] | - | - | - | 74 | - | - | - | - | 2.7 | - |
| Ours (parkour policy) | **86** | **80** | **100** | 73 | 100 | **2.37** | **3.05** | **3.6** | 2.68 | 3.6 |
| Oracles w/o Soft Dyn | 0 | 0 | 93 | 86 | 100 | 1.54 | 1.73 | 3.58 | 1.73 | 3.6 |
| Oracles | 95 | 82 | 100 | 100 | 100 | 3.60 | 3.59 | 3.6 | 2.78 | 3.6 |

Table 2: We test our method against several baselines and ablations in the simulation with a max distance of 3.6m. We measure the success rates and average distances of every skill averaged across 100 trials and 3 random seeds. Our parkour policy shows the best performance using only sensors that are available in the real world. We evaluate on the test environments with obstacles proprieties that are more difficult than the ones of training environments shown in Table 1.

have full knowledge of the type of obstacle related to every state $s_t$, we can assign the corresponding specialized skill policy $\pi_{s_t}^{\text{specialized}}$ to teach the parkour policy how to act at a state. For example, we assign the climb policy $\pi_{\text{climb}}$ to supervise the parkour policy given a high obstacle. We parameterize the policy as a GRU. The inputs except the recurrent latent state are the proprioception $s_t^{\text{proprio}}$, the previous action $a_{t-1}$ and a latent embedding of the depth image $I_t^{\text{depth}}$ processed by a small CNN. The distillation objective is

$$\underset{\theta_{\text{parkour}}}{\arg\min} \mathbb{E}_{s_t, a_t \sim \pi_{\text{parkour}}, sim} \left[ D \left( \pi_{\text{parkour}} \left( s_t^{\text{proprio}}, a_{t-1}, I_t^{\text{depth}} \right), \pi_{s_t}^{\text{specialized}} \left( s_t^{\text{proprio}}, a_{t-1}, e_t^{\text{vis}}, e_t^{\text{phy}} \right) \right) \right],$$

where $\theta_{\text{parkour}}$ are the network parameters of the parkour policy, $sim$ is the simulator with hard dynamics constraints, and $D$ is the divergence function which is binary cross entropy loss for policy networks with tanh as the last layer. Both polices $\pi_{\text{parkour}}$ and $\pi_{s_t}^{\text{specialized}}$ are stateful. More details of the parkour policy network are in the supplementary.

### 3.3 Sim-to-Real and Deployment

Although the distillation training in Section 3.2 can bridge the sim-to-real gap in physical dynamics properties such as terrain friction and mass properties of the robot [8, 10], we still need to address the sim-to-real gap in visual appearance between the rendered depth image in simulation and the onboard depth image taken by a depth camera in the real world. Shown in Figure 5, we apply pre-processing techniques to both the raw rendered depth image and the raw real-world depth image. We apply depth clipping, pixel-level Gaussian noise, and random artifacts to the raw rendered depth image, and apply depth clipping, hole filing, spatial smoothing and temporal smoothing to the raw real-world depth image.

The depth images in both simulation and real-world have a resolution of 48 * 64. Due to the limited onboard computation power, the refresh rate of onboard depth image is 10Hz. Our parkour policy operates at 50Hz in both simulation and the real world to enable agile locomotion skills, and asynchronously fetches the latest latent embedding of the depth image processed by a small CNN. The output actions of the policy are target joint positions which are converted to torques on the order of 1000Hz through a PD controller of $K_p = 50$ and $K_d = 1$. To ensure safe deployment, we apply a torque limits of 25Nm by clipping target joint positions: $\text{clip}(q^{\text{target}}, (K_d * \dot{q} - 25)/K_p + q, (K_d * \dot{q} + 25)/K_p + q)$.

## 4 Experimental Results

**Robot and Simulation Setup.** We use IsaacGym [119] as the simulator to train all the policies. To train the specialized parkour skills, we construct large simulated environments consisting of 40 tracks and 20 obstacles on each track. The obstacles in each track have linearly increasing difficulties based on the obstacle property ranges in Table 1. We use a Unitree A1 and a Unitree Go1 that are equipped with Nvidia Jetson NX for onboard computation and Intel RealSense D435 for onboard visual sensing. More details are in the supplementary.

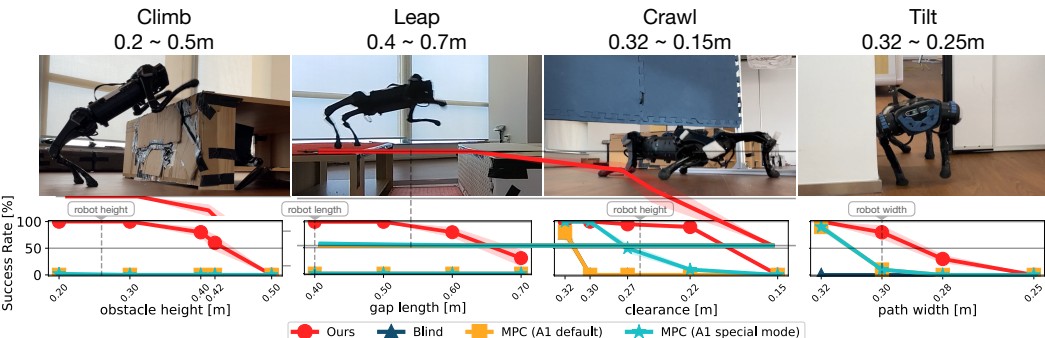

Figure 6: Real-world indoor quantitative experiments. Our parkour policy can achieve the best performance, compared with a blind policy and built-in MPC controllers. We control the MPC in A1 special mode by teleoperating the robot lower down or tilt the body during crawling and tilt respectively.

**Baselines and Ablations.** We compare our parkour policy with several baselines and ablations. The baselines include **Blind**, **RND** [120], **MLP** and **RMA** [8]. The ablations include **No Distill**, **Oracles w/o Soft Dyn**. We also include **Oracles**, specialized parkour skills conditioned on priviledge information in simulation, for the completeness of the comparisons.

- **Blind**: a blind parkour policy baseline distilled from the specialized skills, implemented by setting depth images $I^{\text{depth}}$ as zeros.
- **RND**: a RL exploration baseline method for training specialized skills with bonus rewards based on forward prediction errors. We train it without our RL pre-training on soft dynamics constraints.
- **MLP**: a MLP parkour policy baseline distilled from the specialized skills. Instead of using a GRU, it uses only the depth image, proprioception and previous action at the current time step without any memory to output actions.
- **RMA**: a domain adaptation baseline that distills a parkour policy on a latent space of environment extrinsics instead of the action space.
- **No Distill**: an ablation training a vision-based parkour policy with GRU directly using PPO with our two-stage RL method but but skipping the distillation stage.
- **Oracles w/o Soft Dyn**: an ablation training specialized skill policies using privileged information directly with hard dynamics constraints.
- **Oracles** (w/ Soft Dyn): our specialized skill policies using privileged information trained with our two-stage RL approach.

### 4.1 Simulation Experiments

**Vision is crucial for learning parkour.** We compare the Blind baseline with our approach. Shown in Table 2, without depth sensing and relying only on proprioception, the distilled blind policy cannot complete any climbing, leaping or tilting trials and can only achieve a 13% success rate on crawling. This is expected, as vision enables sensing of the obstacle properties and prepares the robot for execute agile skills while approaching the obstacles.

**RL pre-training with soft dynamics constraints enables parkour skills' learning.** We compare the RND, Oracles w/o Soft Dyn and ours (Oracles w/ Soft Dyn), all trained using privileged information without the distillation stage. We aim to verify that our method of RL pre-training with soft dynamics constraints can perform efficient exploration. In Figure 7, we measure the average success rates of each method averaged over 100 trials across all the parkour skills that require exploration including climbing, leaping, crawling and tilting. We trained using three random seeds for each method to measure the standard deviations. Our method using RL pre-training with soft dynamics constraints can achieve much faster

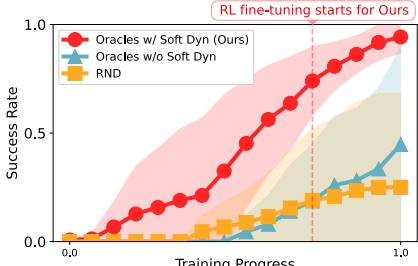

Figure 7: Comparison of specialized oracles trained with soft dynamics constraints with baselines averaged across every skill and three trials.

learning progress and a better final success rate around 95%. We notice that RND struggles to learn meaningful behaviors with scenarios that require fine-grained maneveurs such as crawling through a thin slit, due to its tendency to reach states where future states are difficult to predict. Both RND

and Oracles w/o Soft Dyn cannot make any learning progress on climbing and leaping, the two most difficult parkour skills. More plots showing the success rates for each skill separately are in the supplementary.

**Recurrent networks enable parkour skills requiring memories.** We compare the MLP baseline with ours using a GRU to parameterize the vision-based parkour policy. Shown in Table 2, the MLP baseline cannot learn the climbing and leaping skills and achieve much lower performance on crawling and tilting. Both climbing and leaping requires the robot to hold a short-term memory of the past visual perceptions. For example, during climbing when the robot has its front legs on the obstacles, it still needs memory about the spatial dimensions of the obstacle captured in past depth images to control the rear legs to complete the climbing.

**Distillation is effective for Sim2Real.** We compare the RMA baseline and the No Distill baseline with ours. Although RMA can achieve similar performance on one skill that it is trained on, i.e. tilting, RMA fixes the network parameters of the MLP which processes the latent embeddings of the backbone GRU, and directly copies them from the specialized skill to the distilled policy. Consequently, it cannot distill multiple specialized skill policies, which have different MLP parameters, into one parkour policy. No Distill cannot learn climbing, leaping and tilting due to the complexity of training directly from visual observations without privileged information.

## 4.2  Real-World Experiments

**Emergent Re-trying Behaviors during Climbing.** Our parkour policy has emergent re-trying behaviors in the real world. When trying to overcoming a high obstacle but failing at the first trial, the robot will push itself away from the obstacle to ensure adequate run-up space for subsequent attempts.. Although we do not program such re-trying behaviors, they nicely emerge out of learning with simple rewards. This behavior is also observed in simulation.

**Indoor Quantitative Experiments.** Shown in Figure 1, we test our parkour policy in a constructed parkour terrain consisting of crawling, climbing, and leaping in sequential. We also conduct quantitative indoor experiments in the real world on the A1 robot. In Figure 6, we compare our vision-based parkour policy, with Blind, MPC (A1 default controller) and MPC (A1 special mode). We show the success rates of each method in every skill under varying difficulties averaged over 10 trials each. We change the skill difficulty by modifying the key obstacle properties, such as obstacle heights for climbing and gap length for leaping. In A1 special mode, we directly teleoperate the robot to change its state, such as lowering the body during crawling. We observe that our parkour policy can enable the robot to climb obstacles as high as 0.40m (1.53x robot height) with an 80% success rate, to leap over gaps as large as of 0.60m (1.5x robot length) with an 80% success rate, to crawl beneath barriers as low as of 0.2m (0.76x robot height) with an 90% success rate, and to squeeze through thin slits of 0.28m by tilting (less than the robot width). Our method has the best performance across all skills. Please refer to our  project website for indoor experiment videos.

**Outdoor Experiments.** Shown in Figure 1, we test our robot in the various outdoor environments. We observe that the robot controlled by our parkour policy can complete a wide range of agile parkour skills. It can leap over two disconnected stone stools by the river with a 0.6m wide gap. It can continuously climb several stairs of 0.42m high each. It can crawl beneath a camping cart as well as handle slippery grass terrain. Please refer to our  project website for outdoor experiment videos.

## 5  Conclusion, Limitations and Future Directions

We present a parkour learning system for low-cost robots. We propose a two-stage reinforcement learning method for overcoming difficult exploration problems for learning parkour skills. We also extensively test our system in both simulation and the real world and show that our system has robust performance for various challenging parkour skills in challenging indoor and outdoor environments. However, the current system requires the simulation environments to be manually constructed. As a result, new skills can only be learned when new environments with different obstacles and appearances are added to the simulation. This reduces how atuomatically new skills can be learned. In the future, we hope to leverage recent advances in 3D vision and graphics to construct diverse simulation environments automatically from large-scale real-world data. We will also investigate how we can train agile locomotion skills directly from RGB that contains semantic information instead of depth images.

**Acknowledgments**

We would like to thank Wenxuan Zhou and her Emergent Extrinsic Dexterity project [116] for inspiring our training pipeline allowing penetration. We would also like to thank Xiaozhu Lin, Wenqing Jiang, Fan Nie, Ruihan Yang, Xuxin Chen, Tony Z. Zhao and Unitree Robotics (Yunguo Cui) for their help in the real-world experiments. Zipeng Fu is supported by Stanford Graduate Fellowship (Pierre and Christine Lamond Fellowship). This project is supported by Shanghai Qi Zhi Institute and ONR grant N00014-20-1-2675.

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

# A  Experiment Videos

We perform thorough real-world analysis of our system. Indoor and outdoor experiment videos can be found at https://robot-parkour.github.io.

# B  Details of Training in Simulation

**Specialized Skills.** A specialize skill policy consists of a GRU followed by a MLP that outputs the target joint positions. We concatenate all the observations including proprioception, last action, recurrent latent state of the GRU, privileged visual information and privileged physics information as a flattened vector. It is passed to a one-layer GRU of 256 hidden sizes, followed by an MLP of hidden dimensions of [512, 256, 128]. We use ELU as the activation. The final layer outputs a 12-dimensional vector and be fed to tanh activation function. The action ranges from $-1$ to 1, which is scaled by a constant action scale: 0.4 for hip joints, and 0.6 for thigh and knee joints.

**Rewards, Environments and PPO.** We follow the insights from [10, 8, 35] that use fractal noises to generate terrains, which enforces the foot contact clearance. We use the reward terms for each specialized policies as listed in Table 3 to 7 in the supplementary. We use these parameters to train all five specialized policies, in either RL pre-training with soft dynamics constraint or fine-tuning the with hard dynamics constraints. The key parameters that related to the difficulties of the tasks are shown in the Table 1 of the main paper. Other parameters of the obstacles are set to constants: the obstacles for climbing is 0.8m wide and 0.8m long along the +x direction. The obstacles for the leaping task are gaps of 0.8m wide and 0.8m depth. For crawling, the obstacle is 0.8m wide and 0.3m in the +x direction. For tilting, the length along the +x direction is 0.6m. We use a set of fixed velocity commands $v_x^{\text{target}}$ for each specialized skill during training. We list them in Table 8 of the supplementary. We sample environment randomizations on the robot mass, center of mass of the robot base, motor strength, terrain friction, depth perception latency, camera position, field of view and proprioception delay for each robot during training. The detailed environment randomization parameters are listed in Table 9. The detailed parameters of the PPO algorithm are listed in Table 10 of the supplementary.

Table 3: Reward Scales for Climbing

| Purposes | Hyperparameter Variables | Values |
|---|---|---|
| x velocity | $\alpha_1$ | 1. |
| y velocity | $\alpha_2$ | 1. |
| angular velocity | $\alpha_3$ | 0.1 |
| energy | $\alpha_4$ | $2e-6$ |
| penetration depth | $\alpha_5$ | $1e-2$ |
| penetration volume | $\alpha_6$ | $1e-2$ |

Table 4: Reward Scales for Leaping

| Purposes | Hyperparameter Variables | Values |
|---|---|---|
| x velocity | $\alpha_1$ | 1. |
| y velocity | $\alpha_2$ | 1. |
| angular velocity | $\alpha_3$ | 0.05 |
| energy | $\alpha_4$ | $2e-6$ |
| penetration depth | $\alpha_5$ | $4e-3$ |
| penetration volume | $\alpha_6$ | $4e-3$ |

Table 5: Reward Scales for Crawling

| Purposes | Hyperparameter Variables | Values |
|---|---|---|
| x velocity | $\alpha_1$ | 1. |
| y velocity | $\alpha_2$ | 1. |
| angular velocity | $\alpha_3$ | 0.05 |
| energy | $\alpha_4$ | $2e-5$ |
| penetration depth | $\alpha_5$ | $6e-2$ |
| penetration volume | $\alpha_6$ | $6e-2$ |

Table 6: Reward Scales for Tilting

| Purposes | Hyperparameter Variables | Values |
|---|---|---|
| x velocity | $\alpha_1$ | 1. |
| y velocity | $\alpha_2$ | 1. |
| angular velocity | $\alpha_3$ | 0.05 |
| energy | $\alpha_4$ | $1e-5$ |
| penetration depth | $\alpha_5$ | $3e-3$ |
| penetration volume | $\alpha_6$ | $3e-3$ |

Table 7: Reward Scales for Running

| Purposes | Hyperparameter Variables | Values |
|---|---|---|
| x velocity | $\alpha_1$ | 1. |
| y velocity | $\alpha_2$ | 1. |
| angular velocity | $\alpha_3$ | 0.05 |
| energy | $\alpha_4$ | $1e-5$ |
| penetration depth | $\alpha_5$ | 0. |
| penetration volume | $\alpha_6$ | 0. |

Table 8: Velocity Commands for each Specialized Policy

| Skills | $v_x^{\text{target}}$ (m/s) |
|---|---|
| Running | 0.8 |
| Climbing | 1.2 |
| Leaping | 1.5 |
| Crawling | 0.8 |
| Tilting | 0.5 |

Table 9: Environment Randomizations ($x \pm y$: Gaussian distribution; $[x, y]$: uniform distributions)

| Parameters | Distributions |
|---|---|
| Added Mass | [1.0, 3.0] (kg) |
| Center of Mass (x) | [-0.05, 0.15] (m) |
| Center of Mass (y) | [-0.1, 0.1] (m) |
| Center of Mass (z) | [-0.05, 0.05] (m) |
| Friction Coefficient | [0.5, 1.0] |
| Motor Strength | [0.9, 1.1] |
| Forward Depth Latency | [0.2, 0.26] (s) |
| Camera Position (x) | $0.27 \pm 0.01$ (m) |
| Camera Position (y) | $0.0075 \pm 0.0025$ (m) |
| Camera Position (z) | $0.033 \pm 0.0005$ (m) |
| Camera Pitch | [0.0, 5.0] (deg) |
| Field of View | [85, 88] (deg) |
| Proprioception Latency | [0.0375, 0.0475] (s) |

Table 10: PPO Hyperparameters

| | |
|---|---|
| PPO clip range | 0.2 |
| GAE $\lambda$ | 0.95 |
| Learning rate | 1e-4 |
| Reward discount factor | 0.99 |
| Minimum policy std | 0.2 |
| Number of environments | 4096 |
| Number of environment steps per training batch | 24 |
| Learning epochs per training batch | 5 |
| Number of mini-batches per training batch | 4 |

**Parkour Policy.** The parkour policy consists of a CNN encoder, a GRU and a MLP. The visual embedding from the CNN encoder is concatenated together with the rest of the observation (proprioception, last action and recurrent latent state of the GRU) and fed to the GRU whose output is then processed the MLP module. The detailed parameters of the network structure are listed in Table 11 of the supplementary. An illustration of the parkour training environment in simulation is shown in Figure 8.

Table 11: Parkour Policy structure

| | |
|---|---|
| CNN channels | [16, 32, 32] |
| CNN kernel sizes | [5, 4, 3] |
| CNN pooling layer | MaxPool |
| CNN stride | [2, 2, 1] |
| CNN embedding dims | 128 |
| RNN type | GRU |
| RNN layers | 1 |
| RNN hidden dims | 256 |
| MLP hidden sizes | 512, 256, 128 |
| MLP activation | ELU |

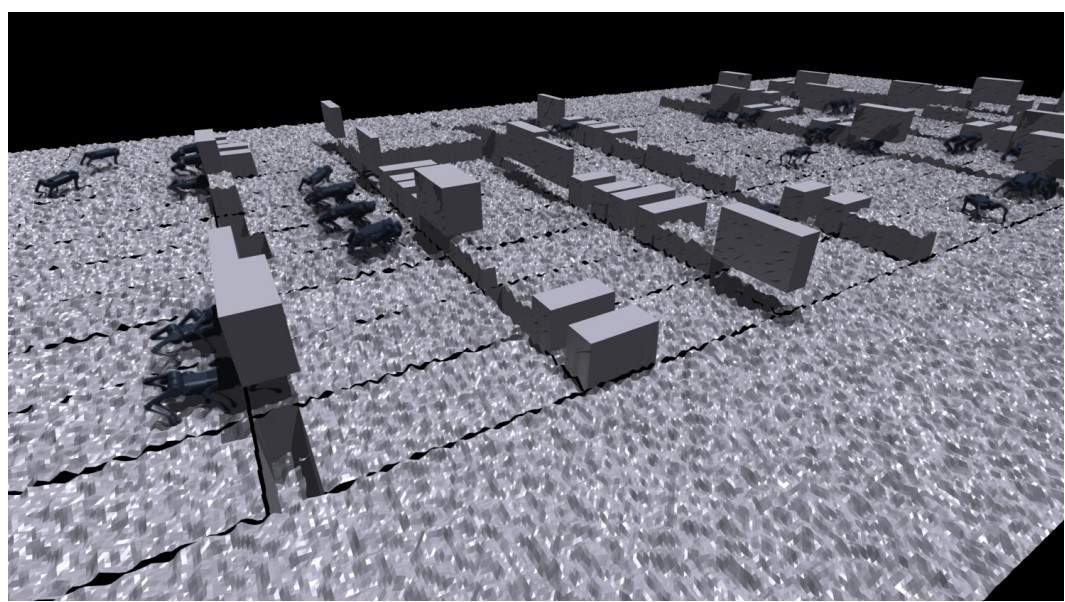

Figure 8: Parkour training environment in simulation during distillation.

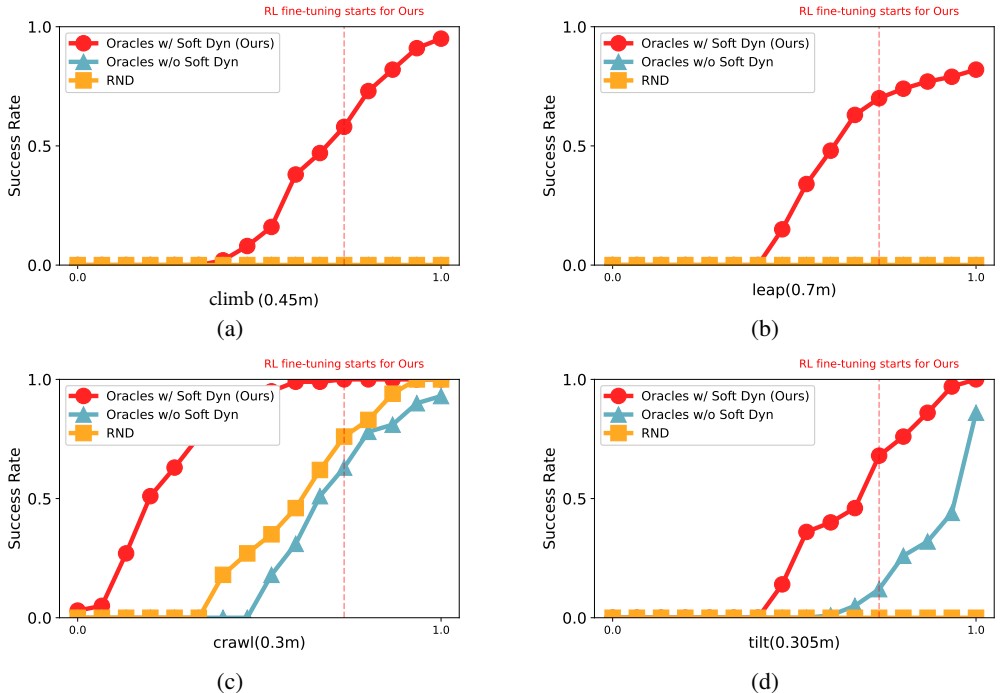

Figure 9: Comparison of our method with Oracles w/o Soft Dyn and RND. For our method, the RL finetuning stage started at the late stages of the training.

We use binary cross-entropy loss for the parkour policy during distillation. The output of both specialized skills and the parkour policy ranges from $-1$ to $1$.

$$D(a^{\text{parkour}}, a^{\text{specialized}}) = \left( \frac{1 + a^{\text{specialized}}}{2} \log \frac{1 + a^{\text{parkour}}}{2} + \frac{1 - a^{\text{specialized}}}{2} \log \frac{1 - a^{\text{parkour}}}{2} \right) \times 2,$$

where $a^{\text{specialized}}$ is the action from the corresponding specialized skill, $a^{\text{parkour}}$ is the action from the parkour policy.

## C   Details of Simulation Setup

We use IsaacGym Preview 4 for simulation. We generate a static large terrain map before each training, during the training of specialized policies. The terrain consists of 800 tracks with a 20 by 40 grid. We set the difficulty of each track in a linear curriculum manner. The tracks in the same row have the same difficulty but differ in non-essential configurations. The tracks in each column are connected end to end so that whenever the robot finished the current track, it keeps moving forward (+x direction) to the more difficult track. We train each specialized policy in soft dynamics using one 1 Nvidia 3090 computer for 12 hours and tune it in hard dynamics for 6 hours. For distillation, we use 4 computers, each of which is equipped with 1 Nvidia 3090 GPU, that share the same NFS file system. We use 3 computers for loading the current training model and collecting the parkour policy's trajectory as well as the specialized policy supervision. We use the other one computer to load the latest trajectories and train the parkour policy.

## D   Details of Robot Setup

We use the Unitree A1 equipped for our real-world experiments which is equipped with an onboard Nvidia Jetson NX. The robot has 12 joints. Each joint is equipped with a motor of 33.5Nm instant maximum torque. It also has a built-in Intel RealSense D435 camera in front of the robot using inferred and stereo to provide depth images. We use ROS1 on Ubuntu 18.04 which runs on the onboard Jetson NX. We use a ROS package based on Unitree SDK to send and receive the robot states as well as the policy command at 100Hz. The ROS package is also equipped with a roll/pitch limit, estimated torque limit, and emergency stop mechanism using the remote control as the means of protection for the robot. To run the policy, we use two Python scripts: a CNN script to run the

visual encoder asynchronously and a main script to run the rest of the networks. We use the Python wrapper of librealsense to capture depth images at the resolution of 240 X 424. We apply the holing filters, spatial filters, and temporal filters from the librealsense utilities. We crop the 60 pixels on the left and 46 pixels on the right before down-sampling the depth image to 48 X 64 resolution. The visual embedding is sent to the main script using ROS message at 10Hz. We fix the policy inference frequency to 50Hz. In each loop, we update the robot proprioception and the visual embedding using ROS and compute the policy output actions. Then we clip the action by a range computed using the current joint position and velocity at a maximum torque of 25Nm, and send the position control command to the ROS package, with $K_p = 50.0, K_d = 1.0$.

## E    Detailed Comparison Studies on RL Pre-Training with Soft Dynamics Constraints

We compare our method with RND and the Oracles w/o Soft Dyn. Our method trained with soft dynamics constraints is the only method that can complete climbing and leaping skills. As shown in Figure 9 of the supplementary, except for crawling, RND fails to learn successful maneuvers to achieve climbing, leaping, and tilting. Although the Oracles w/o Soft Dyn learned to achieve crawl and tilt skills, but fail to learn to climb and leap, which are the most difficult skills among all the skills in this paper.

