# OpenReview forum: "Robot Parkour Learning"
_robot-learning.org/CoRL/2023/Conference — CoRL 2023 Oral_

### Official Review · Reviewer_pRuB · 2023-07-06

**Confidence:** 5
**Originality:** Excellent
**Technical Quality:** Excellent
**Clarity Of Presentation:** Excellent
**Impact:** 4

**Recommendation:**

Strong Accept: I recommend accepting the paper and will argue for my recommendation even if other reviewers hold a different opinion.

**Review:**

This is an absolutely amazing paper, that is impressive technically and experimentally. Well done.

**Quality Of The Limitations Section:**

Limitations are addressed clearly

**Questions For Rebuttal:**

The only question is if the assumptions made (e.g., 5 parkour skills, four types of obstacles, Table 1, etc) affect the extendability and deployment of the methodology.

**Robotics Focus:**

Sufficient demonstration on hardware

**Summary Of Paper:**

This paper introduces an end-to-end parkour system, based on depth sensing. There are 5 policies and 1 simple reward, that are learned via 2-stages RL. The system is deployed on a real robot.

**Summary Of Recommendation:**

A fantastic paper. Of course there are limitations, but compared to the SOTA, this seems like an impressive extension. Well done.

---

### Official Review · Reviewer_PpdX · 2023-07-18

**Confidence:** 5
**Originality:** Very Good
**Technical Quality:** Very Good
**Clarity Of Presentation:** Very Good
**Impact:** 4

**Recommendation:**

Strong Accept: I recommend accepting the paper and will argue for my recommendation even if other reviewers hold a different opinion.

**Review:**

Overall, the paper is excellent. This is a great piece of work, and the results are very impressive. The emergent "retry" behaviour is very inspiring.



**Quality Of The Limitations Section:**

Limitations are addressed clearly

**Questions For Rebuttal:**

The only thing I would ask for the rebuttal proof-read the paper and fix the few inconsistencies. For instance (and there are many other ones). Page 2, we have two sentences starting with "Second, " in a row.
It would also be important to define early on in the paper what "direct Collocation" is. It is not an obvious concept and the paper repeats multiple times that it uses this without telling us what it is.



**Robotics Focus:**

Sufficient demonstration on hardware

**Summary Of Paper:**

This paper presents a method to train end-to-end deep policies to control robots to solve "parkour" tasks. The robot, a quadruped, is now able to jump over obstacles, crawl under bridges, and do many other tasks.
The proposed method relies on 3 training stages. First, soft-constrained training in simulation, second strongly constrained training in simulation, and finally the distillation of 5 different tasks into a single policy. All the training is done in simulation, but proper noise addition the learned policies can transfer directly to the real robot.




**Summary Of Recommendation:**

As I said, this is a great paper that should be presented at CoRL. There is still a bit of polishing to do, but the results and core ideas of the paper are great.

---

### Official Review · Reviewer_u4ib · 2023-07-18

**Confidence:** 4
**Originality:** Good
**Technical Quality:** Fair
**Clarity Of Presentation:** Fair
**Impact:** 3

**Recommendation:**

Weak Accept: I recommend accepting the paper, but will not argue for my recommendation if the majority of other reviewers have a different opinion.

**Review:**

The main strength of this paper is the convincing real-world results. The authors manage to show various experiments in both controlled and uncontrolled settings. The final policy is capable of traversing obstacles where most locomotion controllers (learned or model-based) would fail. While the performance of each skill is not necessarily groundbreaking, the distillation of those skills into a common policy, and the direct use of depth cameras are both interesting directions for research.

On the other hand, this paper could be improved by pushing further the performance of different components of the pipeline. The supplementary video shows that the robot is capable of traversing the obstacles, but it can also be seen that some of the motions are fairly primitive and uncontrolled. The leaping motion seems to move very quickly without clear foot placement leading me to beleive that in many tries it misses the edge of the obstacles. The crawling motions seem to often collide with the obstacle it is avoiding. Finally, in the current state, the tilting motion doesn't seem convincing (it is simply walking through the obstacles with minimal adjustment) and could be removed form the paper.
These visual observations are supported by the reported success rates of the paper. I would encourage the authors to investigate why the leaping skill fails fairly often. Moreover, why does the distilled policy perform significantly worse than the individual skills (climbing failure rate 14% vs 5%). If there are fundamental limitations to the distillation process such as indistinguishable states it should be stated clearly in the paper.

**Quality Of The Limitations Section:**

Additional details required

**Questions For Rebuttal:**

The language of the paper can be significantly improved. The introduction is particularly affected. A spell checker would help find the multiple grammatical errors, typos, and repeated words that are spread across the text. The citations are also overwhelming and unclear in some cases. It is not necessary to cite >20 papers together. Cite other works individually or in small groups (2-3), currently most of the citations are redundant.

The authors claim that their method allows successful training with minimal rewards. However, nothing in the proposed method aims to reduce the reward-tuning process. It seems like the only reason why minimal rewards can be used is that the authors did not spend enough effort trying to achieve smooth and controlled motions. This would prove the opposite of the original claim: more reward tuning is needed to achieve satisfactory results.

It is unclear how the virtual obstacles can lead to the climbing motion. My understanding is that with "soft dynamics" the robot can not interact with the obstacles. As such how can it learn to climb on top of a box? Could the authors show the motions learned at the end of the pre-training stage? Why is this method inspired by direct collocation, after briefly reading the citation the reference is not clear.

Finally, how would the authors generalize their method to cases where obstacles are not placed in a straight line? How would the skill training and distillation processes be affected?



**Robotics Focus:**

Sufficient demonstration on hardware

**Summary Of Paper:**

This paper shows how deep reinforcement learning can be used to achieve parkour-like motions with a small quadrupedal robot. The authors first train different specialized skills before distilling them into a common policy. The paper also proposes to use two-staged training for the skills. First, the robots are trained with "virtual" obstacles, which don't affect the dynamics but lead to penalties when the robot's body intersects them. In the second stage, proper obstacles are added to the simulation, and the skills are fine-tuned. The authors demonstrate that the proposed approach is deployable on the real robot in indoor and outdoor scenarios.

**Summary Of Recommendation:**

This paper tackles the complex task of real-world agile locomotion with an interesting method and shows promising results on hardware.
Nevertheless, these results could be significantly improved without changing the fundamental method.

I believe this paper can be accepted for publication if the authors improve on their results or provide sufficient analysis of the failure cases and describe the corresponding limitations of the approach.

---

### Official Review · Reviewer_rf5K · 2023-07-19

**Confidence:** 5
**Originality:** Fair
**Technical Quality:** Good
**Clarity Of Presentation:** Very Good
**Impact:** 4

**Recommendation:**

Strong Accept: I recommend accepting the paper and will argue for my recommendation even if other reviewers hold a different opinion.

**Review:**

The authors provide impressive real-world experiments, demonstrating robust parkour skills. Additionally, the proposed method of pre-training with soft-dynamic constraints and finetuning with hard-dynamics constraint is surprisingly simple (it can be thought of as a variant of curriculum learning), yet enables emergent parkour skills. I think the paper can be further improved in a few ways:

First, the authors are missing a couple of baselines to compare against. For the comparison against the MLP baseline, many prior works have shown that recurrent networks are useful for tasks that require memory, so the finding that their GRU policy beats MLP is not very novel. Instead, it would be useful to compare against a MLP baseline that receives either a stack of image frames, or history of observations to make it a fair comparison against their method that uses a GRU. Second, the authors are missing a comparison against a mixture-of-experts approach similar to [2], which uses a coordination policy to switch between each expert skill (that has individually been distilled into one that uses vision). It is unclear if distilling the different skill policies to one monolithic policy will lead to degradation of each individual skill, which may be mitigated by a mixture-of-experts approach.

Next, for the simulation and real-world experiments, the authors are missing quantitative results on success rate for the whole parkour task (a sequence of climb, leap, crawl, tilt, in random order). The quantitative results presented (Table 2, Figure 7) only consist of each individual task. Since the claim of the paper is focused on parkour as a whole, it is important to provide results to demonstrate how well the automatic selection of the skill performs, and the success rate of the whole parkour task.

Additionally, it is unclear how well the policy will generalize. Are train and test environments (obstacle sizes, ordering of parkour track) the same? The authors mention the training environment as “40 tracks and 20 obstacles on each track”, what is the testing environment like? For table 2, what are the parameters for the obstacle in each task? Do the authors test against out of distribution obstacle sizes? The obstacles are also all rectangular/box-like, will the robot be able to generalize to different obstacles in the real-world?
The policy currently gets rewarded for forward movement, does this mean that the parkour is limited to a straight line in front of the robot? How can this be extended to a policy that needs to parkour to a given navigation goal that requires turning or path planning?

Lastly, since the method uses only depth images for vision, the robot is missing a lot of semantic information from the environment to decide what skill is best to use. The robot simply needs to know it should crawl if there’s only a small clearing at the bottom, or jump if there is a large obstacle in front of it. What happens if there is a table (with table legs showing an opening at the bottom, as opposed to the platform-like obstacles shown in the real-world experiments)? The authors mention lack of RGB input in passing in the last sentence of the limitations section, but more details on this limitation are necessary. Extending this work to work with RGB images seems non-trivial– sim2real gap is larger, parkour in the outdoors will be harder depending on lighting, shadows, etc, and processing RGB images may lead to larger latencies that may be prohibitive. Could the authors comment on how their work can be extended to overcome these limitations? Is the extension just an engineering effort, or are the authors currently making assumptions that may prohibit this extension?

Minor:
The authors should cite [1], which introduces the idea of learning a policy using privileged information, and distilling this policy into one that doesn’t have access to this privileged information (i.e., a policy that uses onboard sensors).
The figures and tables are often placed too far away from where it is referenced in the text. Please consider re-arranging the ordering to make it easier for the reader to switch between the text / figures.
Please add error bars to table 2.
Typo on line 63, ‘we use DAgger to distill them’
Typo line 270

Originality: The work builds on many tricks and methods known to work in sim2real for locomotion such as: learning by cheating [1], depth sensor filtering [2, 4], automatic curriculum [3], and energy consumption-based reward functions [5].

References:
[1] D. Chen, B. Zhou, V. Koltun, and P. Krahenbuhl, “Learning by cheating,” in Conference on Robot Learning (CoRL), 2019
[2] Yokoyama, Naoki, et al. "Adaptive Skill Coordination for Robotic Mobile Manipulation." arXiv preprint arXiv:2304.00410 (2023).
[3] N. Rudin, D. Hoeller, P. Reist, and M. Hutter. Learning to walk in minutes using massively
parallel deep reinforcement learning. In Conference on Robot Learning, 2022.
[4] Hoeller, David, et al. "Learning a state representation and navigation in cluttered and dynamic environments." IEEE Robotics and Automation Letters 6.3 (2021): 5081-5088.
[5] Z. Fu, A. Kumar, J. Malik, and D. Pathak. Minimizing energy consumption leads to the
403 emergence of gaits in legged robots. In CoRL, 2021.

**Quality Of The Limitations Section:**

Limitations are not well addressed

**Questions For Rebuttal:**

Please address the comments/questions provided in the weaknesses session. It is important to provide additional experiments to the baselines I mentioned above, as well as quantitative experiments for the full parkour task.

**Robotics Focus:**

Sufficient demonstration on hardware

**Summary Of Paper:**

The authors present a method to enable low-cost quadruped robots to perform parkour skills including climbing, leaping, crawling, tilting and running. The key to their proposed method is a two-stage RL method that utilizes pre-training with soft dynamics constraints and finetuning with hard dynamics constraints. The authors demonstrate that their method can perform parkour skills on a real-world robot using onboard visual sensing.

**Summary Of Recommendation:**

The paper currently makes a grand claim that their “parkour policy enables low-cost quadrupedal robots to autonomously select and execute appropriate parkour skill” (L71-72), yet the full parkour task is only shown qualitatively, and the quantitative experiments focus on each parkour skill individually. The additional baselines and experiments referenced in the weaknesses section will improve the claims made by the author.

---

### Decision · Program_Chairs · 2023-08-30

**Decision:**

Accept (Oral)

**Comment:**

The paper presents a novel framework for learning a vision-based parkour policy. Given all the impressive results and extensive analyses, the reviewers converge to strongly support this paper.